# DENSITY ESTIMATION ON LOW-DIMENSIONAL MANI-FOLDS: AN INFLATION-DEFLATION APPROACH

## ABSTRACT

Normalizing Flows (NFs) are universal density estimators based on Neuronal Networks. However, this universality is limited: the density's support needs to be diffeomorphic to a Euclidean space. In this paper, we propose a novel method to overcome this limitation without sacrificing the universality. The proposed method inflates the data manifold by adding noise in the normal space, trains an NF on this inflated manifold and, finally, deflates the learned density. Our main result provides sufficient conditions on the manifold and the specific choice of noise under which the corresponding estimator is exact. Our method has the same computational complexity as NFs, and does not require to compute an inverse flow. We also show that, if the embedding dimension is much larger than the manifold dimension, noise in the normal space can be well approximated by some Gaussian noise. This allows using our method for approximating arbitrary densities on non-flat manifolds provided that the manifold dimension is known.

## 1 INTRODUCTION

Many modern problems involving high dimensional data are formulated probabilistically. Key concepts, such as Bayesian Classification, Denoising or Anomaly Detection, rely on the data generating density $p^*(x)$. Therefore, a main research area and of crucial importance is learning this data generating density $p^*(x)$ from samples.

For the case where the corresponding random variable $X \in \mathbb{R}^D$ takes values on a manifold diffeomorphic to $\mathbb{R}^D$, a Normalizing Flow (NF) can be used to learn $p^*(x)$ exactly (Huang et al., 2018). Recently, a few attempts have been made to overcome this topological constraint. However, to do so, all of these methods either need to know the manifold beforehand (Gemici et al. (2016), Rezende et al. (2020)) or they sacrifice the exactness of the estimate (Cornish et al. (2019), Dupont et al. (2019)).

Our goal in this paper is to overcome both the aforementioned limitations of using NFs for density estimation on Riemannian manifolds. Given data points from a $d-$dimensional Riemannian manifold embedded in $\mathbb{R}^D$, $d < D$, we first inflate the manifold by adding a specific noise in the normal space direction of the manifold, then train an NF on this inflated manifold, and, finally, deflate the trained density by exploiting the choice of noise and the geometry of the manifold. See Figure 1 for a schematic overview of these points.

Our main theorem states sufficient conditions on the manifold and the type of noise we use for the inflation step such that the deflation becomes exact. To guarantee the exactness, we do need to know the manifold as in e.g. Rezende et al. (2020) because we need to be able to sample in the manifold's normal space. However, as we will show, for the special case where $D \gg d$, the usual Gaussian noise is an excellent approximation for a noise in the normal space component. This allows using our method for approximating arbitrary densities on Riemannian manifolds provided that the manifold dimension is known. In addition, our method is based on a single NF without the necessity to invert it. Hence, we don't add any additional complexity to the usual training procedure of NFs.

**Notations:** We denote the determinant of the Gram matrix of $f$ as $g^f(x) := |\det\left(J_f(x)^T J_f(x)\right)|$ where $J_f(x)$ is the Jacobian of $f$. We denote the Lebesque measure in $\mathbb{R}^n$ as $\lambda_n$. Random variables will be denoted with a capital letter, say $X$, and its corresponding state space with the calligraphical

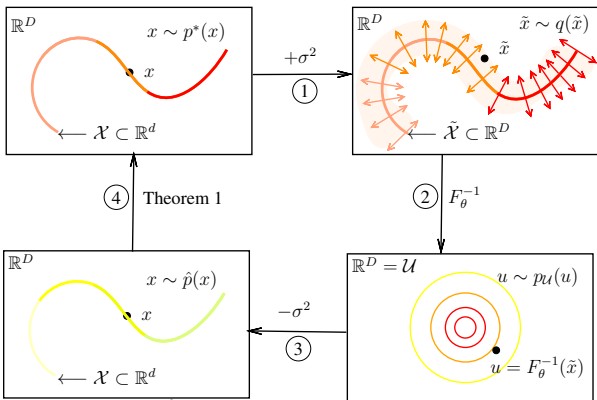

Figure 1: Schematic overview of our method. 1. A density $p^*(x)$ with support on a $d$-dimensional manifold $\mathcal{X}$ (top left) is inflated by adding noise $\sigma^2$ in the normal space (top right). 2. We have an NF $F_\theta^{-1}(x)$ learn this inflated density $q(\tilde{x})$ using a well-known reference measure $p_\mathcal{U}(u)$. 3. We deflate the learned density to obtain an estimate $\hat{p}(x)$ for $p^*(x)$. 4. Our main result provides sufficient conditions for the manifold $\mathcal{X}$ and the choice of noise such that $\hat{p}(x) = p^*(x)$.

version, i.e. $\mathcal{X}$. Small letters correspond to vectors with dimensionality given by context. The letters $d, D, n$, and $N$ are always natural numbers.

## 2 BACKGROUND AND PROBLEM STATEMENT

An NF transforms a known auxiliary random variable by using bijective mappings parametrized by Neuronal Networks such that the given data points are samples from this transformed random variable, see Papamakarios et al. (2019). Formally, an NF is a diffeomorphism $F_\theta : \mathcal{U} \to \mathcal{X}$ and induces a density on $\mathcal{X}$ through $p_\theta(x) = (g^F(u)^{-\frac{1}{2}} p_\mathcal{U}(u)$ where $p_\mathcal{U}(u)$ is known and $u = F_\theta^{-1}(x)$. The parameters $\theta$ are updated such that the KL-divergence between $p^*(x)$ and $p_\theta(x)$,

$$D_{KL}(p^*(x)||p_\theta(x)) = \mathbb{E}_{x \sim p^*(x)}[\log p_\theta(x)] + const. \qquad (1)$$

is minimized. If $F_\theta$ is expressive enough, it was proven that in the limit of infinitely many samples, updating $\theta$ to minimize this objective function converges to a $\theta^*$ such that it holds $\mathbb{P}_X$-almost surely $p^*(x) = p_{\theta^*}(x)$, see (Huang et al., 2018).

More generally, let $X \in \mathcal{X} \subset \mathbb{R}^D$ be generated by an unobserved random variable $Z \in \mathcal{Z} \subset \mathbb{R}^d$ with density $\pi(z)$, that is $X = f(Z)$ for some function $f : \mathcal{Z} \to \mathcal{X}$ where typically $d < D$. In Gemici et al. (2016), $f$ is an embedding[1], and it was shown that one can calculate probabilities such as $\mathbb{P}_X(A)$ for measurable $A \subset \mathcal{X}$ using a density $p^*(x)$ with respect to the volume form $dV_f$ induced by $f$, that is

$$\mathbb{P}(X \in A) = \int_{f^{-1}(A)} \pi(z)dz = \int_A p^*(x)dV_f(x) \qquad (2)$$

with $p^*(x) = \pi(z)g^f(z)^{-\frac{1}{2}}$ and $dV_f(x) = \sqrt{g^f(z)}dz$ where $z = f^{-1}(x)$. Hence, given an explicit mapping $f$ and samples from $p^*(x)$, we can learn the unknown density $\pi(z)$ using a usual NF in $\mathbb{R}^d$. However, in general, the generating function $f$ is either unknown or not an embedding creating numerical instabilities for training inputs close to singularity points.

In Brehmer & Cranmer (2020), $f$ and the unknown density $\pi$ are learned simultaneously. The main idea is to define $f$ as a level set of a usual flow in $\mathbb{R}^D$ and train it together with the flow in $\mathbb{R}^d$ used to learn $\pi(z)$. To evaluate the density, one needs to invert $f$ and thus this approach may be very slow for high-dimensional data. Besides, to guarantee that $f$ learns the manifold they proposed several ad hoc training strategies. We tie in with the idea to use an NF for learning $p^*(x)$ with unknown $f$ and study the following problem.

---

[1]Thus, a regular continuously differentiable mapping (called immersion) which is, restricted to its image, a homeomorphism.

**Problem 1** *Let $\mathcal{X}$ be a $d-$dimensional manifold embedded in $\mathbb{R}^D$. Let $X = f(Z)$ be a random variable generated by an embedding $f : \mathbb{R}^d \to \mathbb{R}^D$ and a random variable $Z \sim \pi(z)$ in $\mathbb{R}^d$. Given $N$ samples from $p^*(x)$ as described above, find an estimator $\hat{p}$ of $p^*$ such that in the limit of infinitely many samples we have that $\hat{p}(x) = p^*(x)$, $\mathbb{P}_X -$ almost surely.*

## 3 METHODS

To solve Problem 1, we want to exploit the universality of NFs. We want to inflate $\mathcal{X}$ such that the inflated manifold $\tilde{\mathcal{X}}$ becomes diffeomorphic to a set $\mathcal{U}$ on which a simple density exists. Doing so, allows us to learn the inflated density $p(\tilde{x})$ exactly using a single NF, see Section 2. Then, given such an estimator for the modified density, we approximate $p^*(x)$ and give sufficient conditions when this approximation is exact.

### 3.1 THE INFLATION STEP

Given a sample $x$ of $X$, if we add some noise $\varepsilon \in \mathbb{R}^D$ to it, the resulting new random variable $\tilde{X} = X + \varepsilon$ has the following density

$$q(\tilde{x}) = \int_{\mathcal{X}} q(\tilde{x}|x) d\mathbb{P}_X(x). \tag{3}$$

Denote the tangent space in $x$ as $T_x$ and the normal space as $N_x$. By definition, $N_x$ is the orthogonal complement of $T_x$. Therefore, we can decompose the noise $\varepsilon$ into its tangent and normal component, $\varepsilon = \varepsilon_{\mathrm{t}} + \varepsilon_{\mathrm{n}}$. In the following, we consider noise in the normal space only, i.e. $\varepsilon_{\mathrm{t}} = 0$, and denote the density of the resulting random variable as $q_{\mathrm{n}}(\tilde{x})$. The corresponding noise density $q_{\mathrm{n}}(\tilde{x}|x)$ has mean $x$ and domain $N_x$. We denote the support of $q_{\mathrm{n}}(\cdot|x)$ by $N_{q_{\mathrm{n}}(\cdot|x)}$. The random variable $\tilde{X} = X + \varepsilon_{\mathrm{n}}$ is now defined on $\tilde{\mathcal{X}} = \bigcup_{x \in \mathcal{X}} N_{q_{\mathrm{n}}(\cdot|x)}$. We want $\tilde{\mathcal{X}}$ to be diffeomorphic to a set $\mathcal{U}$ on which a known density can be defined.

**Example 1** *(a) Let $\mathcal{X} = S^1 = \{x \in \mathbb{R}^2 \mid ||x|| = 1\}$ be the unit circle. For each $x \in S^1$ there exists $z \in [0, 2\pi)$ such that $x = e_r(x) = (\cos(z), \sin(z))^T$. To sample a point $\tilde{x}$ in $N_x$, which is spanned by $e_r(x)$, we sample a scalar value $\gamma$ and set $\tilde{x} = x + \gamma e_r(x)$. With $\gamma \sim Uniform[-1, 1)$, we have that*

$$\tilde{\mathcal{X}} = \bigcup_{x \in \mathcal{X}} \{x + \gamma e_r(x) | \gamma \in [-1, 1)\} = \{x \in \mathbb{R}^2 \mid ||x||_2 < 2\} \tag{4}$$

*which is the open disk with radius $2$. The open disk is diffeomorphic to $(0, 1) \times (0, 1)$. Thus, $q_{\mathrm{n}}(\tilde{x})$ can be learned by a single NF $F^{-1}$ and $p_{\mathcal{U}}(u) = Uniform\left((0, 1) \times (0, 1)\right)$ as reference.*

*(b) As in (a), we consider the unit circle. Now we set $\gamma$ to be a $\chi^2-$ distribution with support $[-1, \infty)$. Then*

$$\tilde{\mathcal{X}} = \bigcup_{x \in \mathcal{X}} \{x + \gamma e_r(x) | \gamma \in [-1, \infty)\} = \mathbb{R}^2. \tag{5}$$

*Thus, $q_{\mathrm{n}}(\tilde{x})$ can be learned by a single NF $F^{-1}$ and $p_{\mathcal{U}}(u) = \mathcal{N}(u; 0, I_D)$ as reference.*

*Both cases can be analogously extended to higher dimensions.*

**Remark 1** *To be precise, the random variable $\varepsilon_{\mathrm{n}}$ is generated by a random variable in $\mathbb{R}^{D-d}$, say $\Gamma$, with measure $\mathbb{P}_\Gamma$. Then, $q_{\mathrm{n}}(\tilde{x}|x)$ is the density of the pushforward of the noise measure $\mathbb{P}_\Gamma$ with regard to the mapping $h : \mathbb{R}^{D-d} \to N_x$. Hence, formally, the density $q(\tilde{x}|x)$ is with respect to the induced volume form $dV_h$, see Section 2. However, if we choose an orthonormal basis for $N_x$, say $n^{(1)}, \ldots, n^{(D-d)}$, then we have that $\tilde{x} = h(\gamma) = A\gamma + x$ where the columns of $A \in \mathbb{R}^{D \times (D-d)}$ are given by these basis vectors, i.e. $A = [n^{(1)}, \ldots, n^{(D-d)}]$. Thus, the Gram determinant of $h$ is $g^h = \det(A^T A) = 1$ and we have that $dV_h(\tilde{x}) = d\gamma$ where $d\gamma$ denotes the volume form with respect to $\lambda_{D-d}$. In this case, we can think of $q_{\mathrm{n}}(\tilde{x}|x)$ as a density with respect to $\lambda_{D-d}$.*

*If $q_{\mathrm{n}}(\tilde{x}|x)$ depends only on $||\tilde{x} - x||$, as it is for the Gaussian distribution, we have that $q_{\mathrm{n}}(\tilde{x}|x) = q_{\mathrm{n}}(||\tilde{x} - x||) = q_{\mathrm{n}}(||\gamma||)$ because $h$ is an isomorphism. Thus, for this case it holds that $q_{\mathrm{n}}(\tilde{x}|x)dV_h(\tilde{x}) = q_{\mathrm{n}}(||\gamma||)d\gamma$. Then, for convenience, we may abuse notation by writing $\gamma \sim q(\tilde{x}|x)$ or $\varepsilon_{\mathrm{n}} \sim r(\gamma)$ where $r(\gamma)$ is the density of $\mathbb{P}_\Gamma$ with respect to $\lambda_{D-d}$.*

## 3.2 THE DEFLATION STEP

Our main idea is to find conditions such that

$$q_{\mathrm{n}}(\tilde{x}) = q_{\mathrm{n}}(\tilde{x}|x)p^*(x) \tag{6}$$

for almost surely all $\tilde{x} \in \tilde{\mathcal{X}}$ and for an almost surely unique $x \in \mathcal{X}$. Because then, given an exact estimator of $q_{\mathrm{n}}(\tilde{x})$, say $\hat{q}_{\mathrm{n}}(\tilde{x})$, we have for $\tilde{x} = x$ that $p^*(x) = \hat{q}_{\mathrm{n}}(x)/q_{\mathrm{n}}(x|x)$.

For equation (6) to be true, we need to guarantee that almost every $\tilde{x}$ corresponds to only one $x \in \mathcal{X}$. This is certainly the case whenever all the normal spaces have no intersections at all (think of a simple line in $\mathbb{R}^2$). We can relax this assumption by allowing null-set intersections. Moreover, only those subsets of the normal spaces are of interest which are generated by the specific choice of noise $q_{\mathrm{n}}(\tilde{x}|x)$. Thus, only the support of $q_{\mathrm{n}}(\tilde{x}|x)$, $N_{q_{\mathrm{n}}(\cdot|x)}$, matters. The key concept for our main result is expressed in the following definition:

**Definition 1** *Let $\mathcal{X}$ be a $d-$dimensional manifold and $N_x$ the normal space in $x \in \mathcal{X}$. Let $q_{\mathrm{n}}(\cdot|x)$ be a density defined on $N_x$ and denote by $N_{q_{\mathrm{n}}(\cdot|x)}$ the domain of $q_{\mathrm{n}}(\cdot|x)$. Denote the collection of all such densities as $Q := \{q_{\mathrm{n}}(\cdot|x)\}_{x\in\mathcal{X}}$. For $\tilde{x} \in \tilde{\mathcal{X}}$, we define the set of all possible generators of $\tilde{x}$ as $\mathcal{A}(\tilde{x}) = \{x' \in \mathcal{X}|N_{q_{\mathrm{n}}(\cdot|x')} \ni \tilde{x}\}$. We say $\mathcal{X}$ is $Q-$**normally separated** if for all $x \in \mathcal{X}$ holds that $\mathbb{P}_{\tilde{X}|X=x}[\tilde{x} \in N_x|\#\mathcal{A}(\tilde{x}) > 1] = 0$ where $\#\mathcal{A}(\tilde{x})$ is the cardinality of the set $\mathcal{A}(\tilde{x})$. In words, every $\tilde{x} \in N_x$ is $\mathbb{P}_{\tilde{X}|X=x}$-almost surely determined by $x$.*

To familiarize with this concept, consider Figure 2 and the following example:

**Example 2** *For the circle in example 1, we choose $\varepsilon_{\mathrm{n}}$ to be uniformly distributed on the half-open interval $[-1, 1)$. The point $(0,0)^T$ is contained in $N_{q_{\mathrm{n}}(\cdot|x)}$ for all $x \in \mathcal{X}$ and thus $N_{q_{\mathrm{n}}(\cdot|x')} \cap N_{q_{\mathrm{n}}(\cdot|x)} = \{(0,0)^T\}$ for all $x \neq x'$, see Figure 2 (middle). Hence, for any given $\tilde{x} \in N_x$ we have that $\mathcal{A}(\tilde{x}) = \begin{cases} \mathcal{X} & \text{if } \tilde{x} = (0,0)^T, \\ x & \text{else,} \end{cases}$ and therefore $\#\mathcal{A}(\tilde{x}) = \begin{cases} \infty & \text{if } \tilde{x} = (0,0)^T, \\ 1 & \text{else.} \end{cases}$*

*Thus, $\mathbb{P}_{\tilde{X}|X=x}\left[\tilde{x} \in \tilde{\mathcal{X}}|\#\mathcal{A}(\tilde{x}) > 1\right] = \mathbb{P}_{\tilde{X}|X=x}\left[\tilde{x} = (0,0)^T\right] = 0$ for all $x \in \mathcal{X}$. What follows is that $\mathcal{X}$ is $Q-$normally seperated.*

*If we were to choose $\varepsilon_{\mathrm{n}}$ to be uniformly distributed on $[-1.5, 1)$, see Figure 2 (right), the normal spaces would overlap and we would have that $\mathbb{P}_{\tilde{X}|X=x}\left[\tilde{x} \in \tilde{\mathcal{X}}|\#\mathcal{A}(\tilde{x}) > 1\right] > 0$. In this case, $\mathcal{X}$ would not be $Q-$normally seperated.*

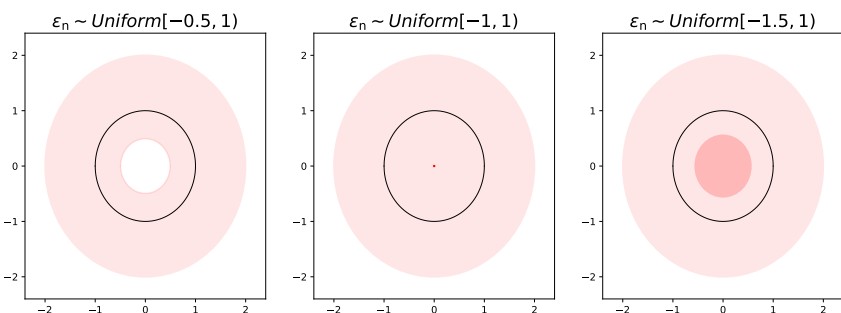

Figure 2: Q-normal separability for different noise distributions $q_{\mathrm{n}}(\tilde{x}|x)$ used to inflate $\mathcal{X} = S^1$ (black line). **Left:** $\mathcal{X}$ is Q-normally separable since every point in the inflated space $\tilde{\mathcal{X}}$ (red shaded area) has a unique generator. **Middle:** $\mathcal{X}$ is Q-normally separable since $\mathbb{P}_{\tilde{X}}-$almost every point in $\tilde{\mathcal{X}}$ has a unique generator. **Right:** $\mathcal{X}$ is not Q-normally separable since every point in the dark shaded area has two generators.

**Theorem 1** *Let $\mathcal{X}$ be a $d-$dimensional manifold. For each $x \in \mathcal{X}$, let $q_{\mathrm{n}}(\cdot|x)$ denote a distribution on the normal space of $x$. Let $\mathcal{X}$ be $Q-$normally separated where $Q := \{q_{\mathrm{n}}(\cdot|x)\}_{x\in\mathcal{X}}$. Assume*

*that we can learn the density $q_\mathrm{n}(\tilde{x})$ as defined in equation (3), by using a single NF $F^{-1}$, thus $q_\mathrm{n}(\tilde{x}) = (g^F(F^{-1}(\tilde{x})))^{-\frac{1}{2}} p_\mathcal{U}(F^{-1}(\tilde{x}))$ for some known density $p_\mathcal{U}$. Then, for $\mathbb{P}_{\tilde{X}}-$almost all $\tilde{x} \in \widetilde{\mathcal{X}}$ holds that $q_\mathrm{n}(\tilde{x}) = p^*(x) q_\mathrm{n}(\tilde{x}|x)$, therefore this equation, when evaluated at $\tilde{x} = x$, yields*

$$p^*(x) = \frac{q_\mathrm{n}(x)}{q_\mathrm{n}(x|x)}. \tag{7}$$

The proof can be found in Appendix A.1.

### 3.3 Gaussian noise as normal noise and the choice of $\sigma^2$

Our proposed method depends on three critical points. First, we need to be able to sample in the normal space of $\mathcal{X}$. Second, we need to determine the magnitude and type of noise. Third, we need to make sure that the conditions of Theorem 1 are fulfilled. We address (partially) those three points.

*1.* For the special case where $D \gg d$, we show that a full Gaussian noise is an excellent approximation for a Gaussian noise restricted to the normal space. Consider $\varepsilon = \varepsilon_\mathrm{t} + \varepsilon_\mathrm{n}$, $\varepsilon \sim \mathcal{N}(0, \sigma^2 I_D)$. Then, the expected absolute squared error when approximating normal noise with full Gaussian noise is $\mathbb{E}\left[|\varepsilon - \varepsilon_\mathrm{n}|^2\right] = \mathbb{E}\left[|\varepsilon_\mathrm{t}|^2\right] = d\sigma^2$. The expected relative squared error is therefore

$$\mathbb{E}\left[\frac{|\varepsilon - \varepsilon_\mathrm{n}|^2}{|\varepsilon_\mathrm{n}|^2}\right] = \mathbb{E}\left[\frac{|\varepsilon_\mathrm{t}|^2}{|\varepsilon_\mathrm{n}|^2}\right] = d\sigma^2 \mathbb{E}\left[\frac{1}{|\varepsilon_\mathrm{n}|^2}\right] = \frac{d}{D-d-2} \tag{8}$$

because $\varepsilon_\mathrm{t}$ and $\varepsilon_\mathrm{n}$ are independent and $\frac{D-d}{|\varepsilon_\mathrm{n}|^2}$ follows a scaled inverse $\chi^2-$distribution with $D - d$ degrees of freedom and scale parameter $1/\sigma^2$. Thus, if $D \gg d$, Gaussian noise is an excellent approximation for a Gaussian in the normal space. We denote the inflated density with Gaussian noise by $q_{\sigma^2}(\tilde{x})$ in the following.

*2.* The inflation must not garble the manifold too much. For instance, adding Gaussian noise with magnitude $\sigma \geq r$ to $S^1$ will blur the circle. Since the curvature of the circle is $1/r$, intuitively, we want $\sigma$ to scale with the second derivative of the generating function $f$. Additionally, we do not want to lose the information of $p^*(x)$ by inflating the manifold. If the generating distribution $\pi(z)$ makes a sharp transition at $z_0$, $\pi(z_0 - \Delta z_o) \ll \pi(z_0 + \Delta z_o)$ for $|\Delta z_o| \ll 1$, adding to much noise in $x_0 = f(z_0)$ will smooth out that transition. Hence, we want $\sigma$ to inversely scale with $\pi''(z)$. We formalize these intuitions in Proposition 1 and prove them in Appendix A.2. In accordance with Theorem 1, we say $p_\sigma(\tilde{x})$ approximates well $p^*(x)$ if $\lim_{\sigma^2 \to 0} p_\sigma(x)/q_\mathrm{n}(x|x) = p^*(x)$ for all $x \in \mathcal{X}$ where $q_\mathrm{n}(x|x)$ is the normalization constant of a $(D-d)-$dimensional Gaussian distribution.

**Proposition 1** *Let $X \in \mathbb{R}^D$ be generated by $Z \sim \pi(z)$ through an embedding $f : \mathbb{R}^d \to \mathbb{R}^D$, i.e. $f(Z) = X$. Let $\pi \in C^2(\mathbb{R}^d)$. For $q_{\sigma^2}(\tilde{x})$ to approximate well $p^*(x)$, in the sense that $\lim_{\sigma^2 \to 0} q_{\sigma^2}(x)/q_\mathrm{n}(x|x) = p^*(x)$ for $x \in \mathcal{X}$, a necessary condition is that:*

$$\left|\frac{\sigma^2}{2\pi(z_0)}||\pi''(z_0) \odot (J_f^T J_f)^{-1}||_+\right| \ll 1 \tag{9}$$

*where $||A||_+ = \sum_{i,j=1}^d A_{ij}$ for $A \in \mathbb{R}^d \times \mathbb{R}^d$ and $\odot$ denotes the elementwise product, and $(\pi''(z_0))_{ij} = \frac{\partial^2 \pi(z)}{\partial z_i \partial z_j}|_{z=z_0}$ is the Hessian of the prior evaluated at $z_0 = f^{-1}(x)$.*

Intuitively, a second necessary condition is that the noise magnitude should be much smaller than the radius of the curvature of the manifold which directly depends on the second-order derivatives of $f$. This can be illustrated in the following example:

**Example 3** *For the circle[2] in $\mathbb{R}^2$ generated by $f(z) = (\cos(z), \sin(z))^T$ and a von Mises distribution $\pi(z) \propto \exp(\kappa \cos(z))$, we get that $\sigma^2 \ll \min\left(\left|\frac{2r^2}{\kappa(\kappa \sin^2(z) - \cos(z))}\right|, r^2\right)$ where the first condition comes from Proposition 1 and the second one comes from the curvature argument.*

Even though this bound may not be usefull as such in practice when $f$ and $\pi$ are unknown, it can still be used if $f$ and $\pi$ are estimated locally with nearest neighbor statistics.

---

[2]Technically, the circle does not fulfill the conditions of Proposition 1 since the domain of $f$ is not $\mathbb{R}$.

From a numerical perspective, inflating a manifold using Gaussian noise circumvents degeneracy problems when training an vanilla NF for low-dimensional manifolds. In particular, the flows Jacobian determinant becomes numerically unstable, see equation (1). This determinant is essentially a volume changing factor for balls. From a sampling perspective, these volumes can be estimated with the number of samples falling into the ball divided by the total number of points. Therefore, we suggest to lower bound $\sigma$ with the average nearest neighbor obtained from the training set to make sure that these volumes are not empty and thus avoid numerical instabilities.

*3.* Intuitively, if the curvature of the manifold is not too high and if the manifold is not too entangled, $Q-$normal separability is satisfied for a sufficiently small magnitude of noise. Also in the manifold learning literature, the entangling must not be too strong. Informally, the reach number provides a necessary condition on the manifold such that it is learnable through samples, see Chapter 2.3 in Berenfeld & Hoffmann (2019). Formally, the reach number is the maximum distance $\tau_{\mathcal{X}}$ such that for all $\tilde{x}$ in a $\tau_{\mathcal{X}}-$neighbourhood of $\mathcal{X}$ the projection onto $\mathcal{X}$ is unique. In Appendix A.3 we prove Theorem 2 which states that any closed manifold $\mathcal{X}$ with $\tau_{\mathcal{X}} > 0$ is $Q-$normally separable.

**Theorem 2** *Let $\mathcal{X} \subset \mathbb{R}^D$ be a closed d-dimensional manifold. If $\mathcal{X}$ has a positive reach number $\tau_{\mathcal{X}}$, then $\mathcal{X}$ is $Q-$normally separable where $Q := \{q_{\mathrm{n}}(\cdot|x)\}_{x \in \mathcal{X}}$ is the collection of uniform distributions on a ball with radius $\tau_{\mathcal{X}}$, i.e. $q_{\mathrm{n}}(\tilde{x}|x) = Uniform(\tilde{x}; B(x, \tau_{\mathcal{X}}) \cap N_x)$ where $B(x, \tau_{\mathcal{X}})$ denotes a $D-$dimensional ball with radius $\tau_{\mathcal{X}}$ and center $x$.*

## 4 RESULTS

We have three goals in this section: First, we numerically confirm the scaling factor in equation (7). Second, we verify that Gaussian noise can be used to approximate a Gaussian noise restricted to the normal space. Third, we numerically test our bound for $\sigma^2$ derived in Section 3.3. Finally, we show that we can learn complicated distributions on $S^2$ without using explicit charts. For training details, we refer to Appendix B.1 and B.2, respectively.

### 4.1 VON MISES ON A CIRCLE

Let $\mathcal{X}$ be a circle with radius 3 and let $\pi(z) \propto \exp(8\cos(z))$ be a $1D$ von Mises distribution. Given $z \sim \pi(z)$, we generate a point in $\mathcal{X}$ according to the mapping $f(z) = 3(\cos(z), \sin(z))$. We want to learn the induced density $p^*(x)$. Note that $3p^*(x) = \pi(z)$ since $1/3$ is the volume form induced by $f$. To benchmark our performance, we use the idea in Gemici et al. (2016) to first embed the circle into $\mathbb{R}$, using e.g. $f^{-1}$, learn the density there with an NF, and transform this learned density back to $S^1$. In Brehmer & Cranmer (2020), this method is named Flow on manifolds (FOM) and we stick to this notation in the following. Note that $f$ is not injective and to illustrate the benefit of our method we choose the singularity point to be $(1, 0)^T$. By moving points close to $(1, 0)^T$ slightly away from $(1, 0)^T$, we numerically ensure that $f$ is an embedding.

**1. The Inflation step:** We inflate $\mathcal{X}$ using 3 types of noise: Gaussian in the normal space (NG), Gaussian in the full ambient space (FG), and $\chi_2$-noise as described in examples 1(b) with scale parameter 3. Technically, NG violates the $Q-$normal separability assumption. However, if $\sigma^2$ is small and the scale parameter for the von Mises distribution is large enough, this is practically fulfilled.

**2. Learning $q_{\mathrm{n}}(\tilde{x})$:** For the NFs we use a Block Neural Autoregressive Flow (BNAF) (De Cao et al., 2019). We use the same NF architecture and training procedures across all models.

**3. Deflation:** Given an estimator for $q_{\mathrm{n}}(\tilde{x})$, we use equation (7) to calculate $p^*(x)$. For FG and NG, we have that $q_{\mathrm{n}}(x|x) = 1/\sqrt{2\pi\sigma^2}$ and for the normal $\chi^2-$noise is $q_{\mathrm{n}}(x|x) = \sqrt{3}e^{-3/2}/(\sqrt{8}\Gamma(\frac{3}{2}))$.

In Figure 3, we show the results for $\sigma^2 = 0.01$ and $\sigma^2 = 1$. In the respective plot, the first row shows training samples from the inflated distributions $q_{\sigma^2}(\tilde{x})$ (left), and $q_{\mathrm{n}}(\tilde{x})$ (middle), respectively. We color code a sample $\tilde{x} = x + \varepsilon$ according to $p^*(x)$ to illustrate the impact of noise on the inflated density. Note that the FOM model (top right) does not need any inflation and therefore is trained on samples from $p^*(x)$ only. In the respective plot, the second row shows the learned density for the different models and compares it to the ground truth von Mises distribution $\pi(z)$ depicted in black. As we can see, for $\sigma^2 = 0.01$ all models perform very well, although the FOM model

slightly fails to capture $p(z)$ for $z$ close to 0 which corresponds to the chosen singularity point. For $\sigma^2 = 1$, we see a significant drop in the performance of the Gaussian model. Although the manifold is significantly disturbed, the normal noise model still learns the density almost perfectly[3], so does the normal $\chi^2-$noise model, as predicted by Theorem 1.

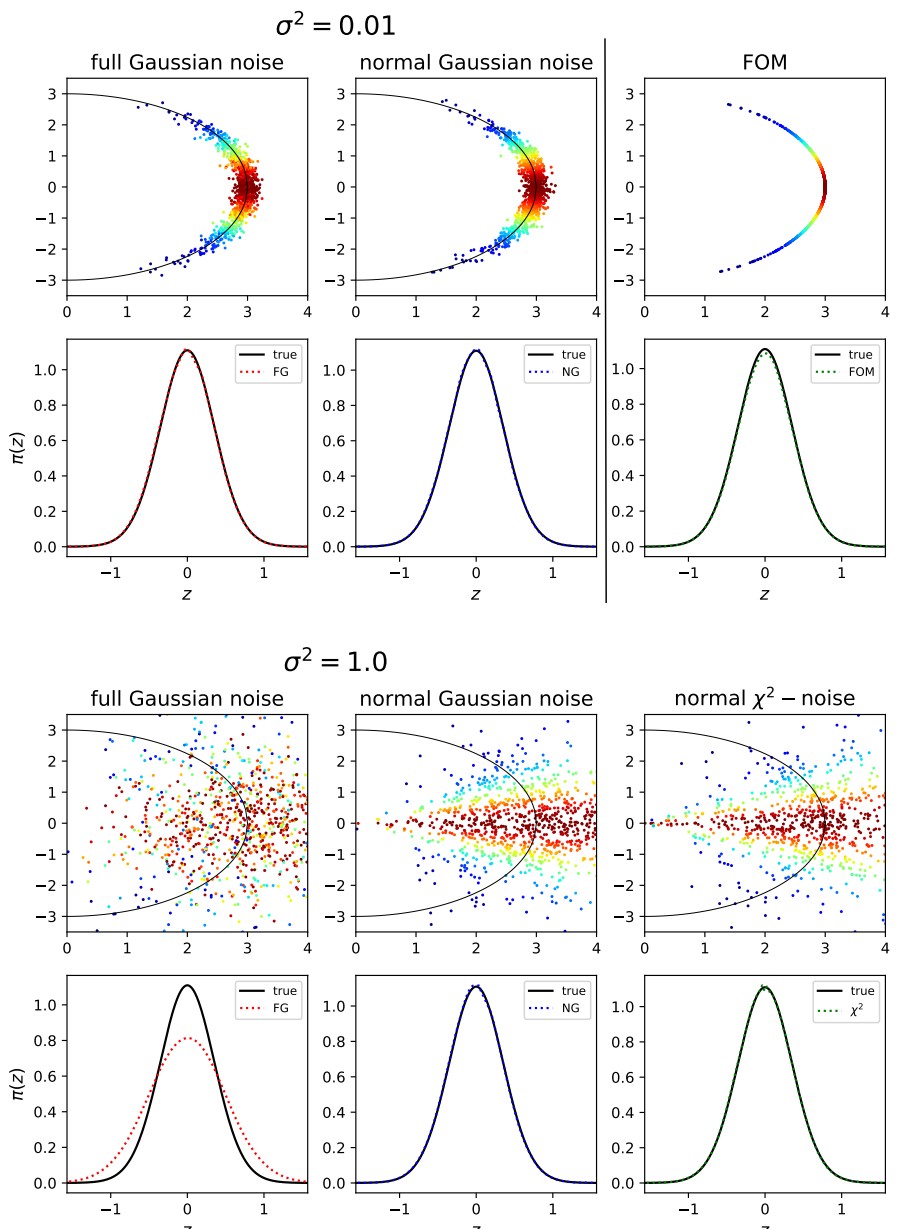

Figure 3: Learned densities for $\sigma^2 = 0.01$ (above) and $\sigma^2 = 1$ (below), respectively. **First row:** Samples used for training the respective model: FG (left), NG (middle), FOM/ $\chi^2$ (right). The black line depicts the manifold $\mathcal{X}$ (a circle with radius 3) and the color codes the value of $p^*(x)$. **Second row:** Colored line: Learned density $\hat{p}(x)$ according to equation (7) multiplied by 3. Blackline: ground truth von Mises distribution.

To measure the dependence of our method on the magnitude of noise, we iterate this experiment for various values of $\sigma^2$ and estimate the Kolmogorov-Smirnov (KS) statistics. The KS statistic is

---

[3]Note that our method still depends on how well an NF can learn the inflated distribution.

defined as $KS = \sup_{x \in \mathcal{X}} |F(x) - G(x)|$, where $F$ and $G$ are the cumulative distribution functions associated with the probability densities $p(x)$ and $q(x)$, respectively. By definition, $KS \in [0, 1]$ and $KS = 0$ if and only if $p(x) = q(x)$ for almost every $x \in \mathcal{X}$. However, equation (6) is only valid if the conditions of Theorem 1 are fulfilled. There is no reason why using equation (7) for the full Gaussian noise would lead to a density on the manifold $\mathcal{X}$. The KS statistics is ill-posed in this case. Nevertheless, we are interested in measuring the sensitivity to the noise, and thus consider the KS statistics as a relative performance measure.

In Figure 4, we display the KS values depending on different levels of noise, for the NG (blue) and FG (orange) noise compared with the ground truth von Mises distribution. Also, we embed the circle into higher dimensions $D = 5, 10, 15, 20$ and repeat this experiment. The result for $D = 2$ and $D = 20$ are shown in the first row (left and right).[4] For $D = 2$, we add the FOM model (which is independent of $\sigma^2$) horizontally. Besides, we depict the lower and upper bound for $\sigma^2$ from Chapter 3.3 with dashed vertical lines. In the lower-left image, we show the optimal $KS$ values obtained for both models depending on $D$. The lower-right image shows the corresponding $\sigma^2$ for those optimal $KS$. In bright, the optimal average $\sigma^2$ is shown whereas the dark regions are the minimum respectively maximum values for $\sigma^2$ such that we outperformed the FOM benchmark. We note that for both cases, the averaged optimal $\sigma^2$ is within the predicted bounds for $\sigma^2$.

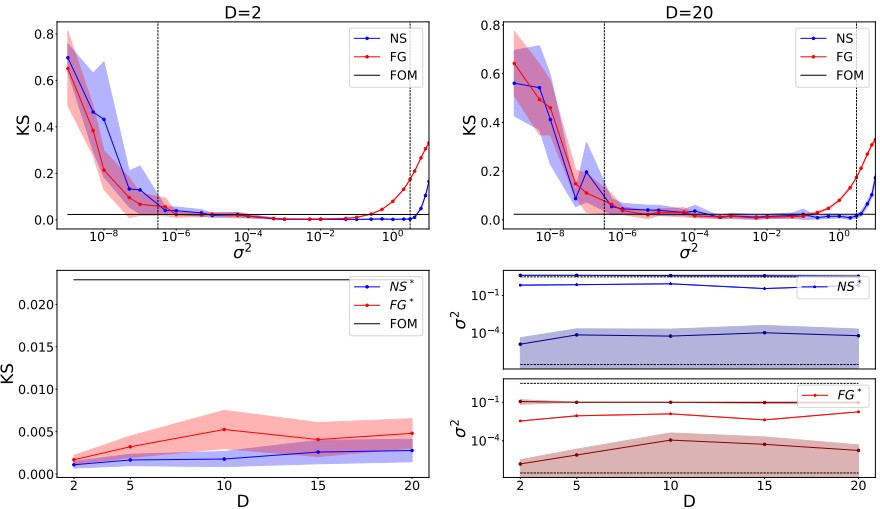

Figure 4: KS values for the NG- (**blue**) and FG-noise method (**orange**) depending on $\sigma^2 \in [10^{-9}, 10]$ and the embedding dimension $D = 5, 10, 15, 20$ in log-scale. For $D = 2$ and $D = 20$ (top right), the two vertical lines represent the lower and upper bound for $\sigma^2$ estimated according to Chapter 3.3 with 10K samples. We plot horizontally the KS value obtained from FOM. **Bottom left:** Optimal KS values depending on $D$. **Bottom right:** Optimal averaged $\sigma^2$ such that optimal $KS$ is obtained (bright). The maximum and minimum $\sigma^2$ such that the FOM benchmark is outperformed (dark). The dashed horizontal lines are again the theoretical bounds. We used 10 seeds for the error bars and plot in log-scale.

Several aspects are remarkable. The flat course of the KS vs. $\sigma^2$ plot is an indicator that the method is not very sensitive to noise and this does not change with the dimensionality of the embedding space. Also, the optimal KS values do not change much depending on $D$ and the NS and FG model approach each other, as predicted.

Interestingly, the onset for the increase in the KS value for the NS-noise is roughly 3 which is the radius of the circle. For increasing $\sigma^2$, $\tilde{\mathcal{X}}$ resembles more and more a double cone which is not

---

[4]Note that the scaling factor depends on $D$, $q_n(x|x) = 1/(2\pi\sigma^2)^{\frac{D-d}{2}}$.

diffeomorphic to $\mathbb{R}^2$ and thus the NF used to train the inflated distribution may not be able to capture the density close to the circle's center correctly.

## 4.2   MIXTURES ON $S^2$

We show that we can learn a complicated distribution, a mixture of 2-dimensional von Mises distributions on a sphere with radius 1, without using any knowledge about the manifold except for its intrinsic dimension. For certain magnitudes of $\sigma^2$, we obtain similar estimates as the FOM benchmark as we can see in the direct comparison of the learned densities, see Figure 5 (top right), and the KS statistics (bottom right). As for the circle, the Gaussian restricted in the normal space allows for a greater range of noise magnitude without sacrificing the quality of the estimate.

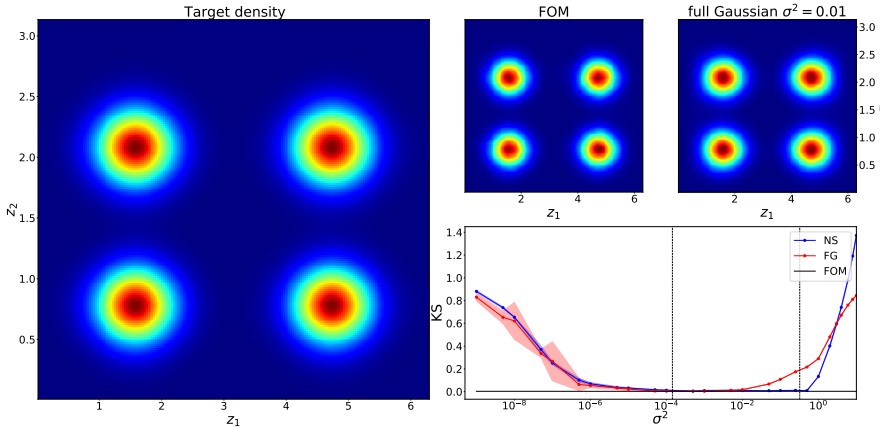

Figure 5: **Left:** Target density. **Upper right:** Learned densities using FOM and our method with Gaussian noise and $\sigma^2 = 0.01$. **Lower right:** KS vs. $\sigma^2$ plot of the Gaussian noise model (full and in normal space) compared to the FOM with the theoretical bounds from Chapter 3.3 for $\sigma^2$ depicted in vertical dashed lines (with 10K samples used to approximate these bounds).

## 5   DISCUSSION

To overcome the limitations of NFs to learn a density $p^*(x)$ defined on a low-dimensional manifold, we proposed to embed the manifold into the ambient space such that it becomes diffeomorphic to $\mathbb{R}^D$, learn this inflated density using an NF, and, finally, deflate the inflated density according to Theorem 1. There, we provided sufficient conditions on the choice of inflation such that we can compute $p^*(x)$ exactly. Notably, we don't need to assume that $p^*(x)$ is supported on a flat manifold. Our method depends on some critical points which we addressed in Section 3.3. So far, the magnitude of noise $\sigma^2$ when using NFs on real-world data is somewhat chosen arbitrarily. As a first step to overcome this arbitrariness, we derived an upper bound for $\sigma^2$ in Proposition 1 and established an interesting connection to the manifold learning literature in Theorem 2.

We hope that our theoretical results motivate some new research directions. Using full Gaussian noise to learn the inflated distribution smears information on $p^*(x)$, in particular if $p^*(x)$ has many local extrema. This loss of information may be especially impactful in out of distribution (OOD) detection or when it comes to adversarial robustness. Therefore, developing methods which allow to generate noise in the manifold's normal space could improve the performance of NFs on such tasks. Another interesting direction is to exploit the product form of equation (6) and learn low-dimensional representations by forcing the NF to be noise insensitive in the first $d$-components and noise sensitive in the remaining ones. Inverting the corresponding flow allows to sample directly from the manifold which has the potential to improve the generative ability of NFs.

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

## A  APPENDIX

### A.1  PROOF OF THEOREM 1

We denote the probability measure of the random variable $X$ as $\mathbb{P}_X$ and it is defined on $(\mathcal{X}, \mathcal{B}(\mathcal{X}))$ where $\mathcal{B}(\mathcal{X})$ is the set of borel measure in $\mathbb{R}^D$ intersected with $\mathcal{X}$. For a realisation of $X$, say $x$, we denote the probability measure of the shifted random variable $x + \varepsilon_n$ as $\mathbb{P}_{\tilde{X}|X=x}$ and it is defined on $(\mathcal{N}_x, \mathcal{B}(\mathcal{N}_x))$. We extend both measures to $(\mathbb{R}^D, \mathcal{B}(\mathbb{R}^D))$ by setting the probabilities to 0 whenever a set $A \in \mathcal{B}(\mathbb{R}^D)$ has no intersection with $\mathcal{X}$ or $N_x$, respectively. For instance, that means for $\tilde{x} \in N_x$ that

$$\mathbb{P}[x + \varepsilon_n \in (\tilde{x}, \tilde{x} + d\tilde{x})] = \mathbb{P}[x + \varepsilon_n \in (\tilde{x}, \tilde{x} + d\tilde{x}) \cap N_x] = \mathbb{P}_{\tilde{X}|X=x}[(\tilde{x}, \tilde{x} + d\tilde{x}) \cap N_x] \quad (10)$$

where $(\tilde{x}, \tilde{x} + d\tilde{x})$ denotes an infinitesimal volume element around $\tilde{x}$.

The mapping $(x, \epsilon) \mapsto x + \epsilon$ is $\mathcal{B}(\mathbb{R}^D) \times \mathcal{B}(\mathbb{R}^D)$−measurable because $\mathbb{R}^D$ is a topological vector space and $\mathcal{B}(\mathbb{R}^D)$ the Borel $\sigma$−algebra. What follows is that $\tilde{X} = X + \varepsilon_n$ is a random variable on $(\mathbb{R}^D, \mathcal{B}(\mathbb{R}^D))$ and has the pushforward of $\mathbb{P}_{(X,\varepsilon_n)}$ with regard to the mapping $(x, \varepsilon_n) \to x + \varepsilon_n$ as probability measure where $\mathbb{P}_{(X,\varepsilon_n)}$ is the joint measure of $X$ and $\varepsilon_n$. Thus, for $A \in \mathcal{B}(\widetilde{\mathcal{X}})$, we have that

$$\mathbb{P}_{\tilde{X}}(A) = \mathbb{P}_{(X,\varepsilon_n)}\left(\{(x, \epsilon) \in \mathbb{R}^D \times \mathbb{R}^D | x + \epsilon \in A\}\right). \quad (11)$$

Now let $\tilde{x} \in N_x$ for an $x \in \mathcal{X}$. Since $\mathcal{X}$ is $Q-$normally separated, $\mathbb{P}_{\tilde{X}}-$almost all $\tilde{x}$ are uniquely determined by $(x, \epsilon)$ such that $\tilde{x} = x + \epsilon$. Therefore, we have for $\mathbb{P}_{\tilde{X}}-$almost all $\tilde{x} = x + \epsilon$ that

$$
\begin{aligned}
\mathbb{P}_{\tilde{X}}((\tilde{x}, \tilde{x} + d\tilde{x}) \cap \widetilde{\mathcal{X}}) &= \mathbb{P}_{(X, \varepsilon_n)}\left(\{(x, \epsilon) \in \mathbb{R}^D \times \mathbb{R}^D | x + \epsilon \in (\tilde{x}, \tilde{x} + d\tilde{x}) \cap \widetilde{\mathcal{X}}\}\right) \\
&= \mathbb{P}\left(X + \varepsilon_n \in (\tilde{x}, \tilde{x} + d\tilde{x}) \cap \widetilde{\mathcal{X}}\right) \\
&= \mathbb{P}\left(X \in (x, x + dx) \cap \mathcal{X}\right) \cdot \mathbb{P}\left(x + \varepsilon_n \in (\tilde{x}, \tilde{x} + d\tilde{x}) \cap N_x\right) \\
&= \mathbb{P}_X\left((x, x + dx) \cap \mathcal{X}\right) \cdot \mathbb{P}_{\tilde{X}|X=x}\left((\tilde{x}, \tilde{x} + d\tilde{x}) \cap N_x\right)
\end{aligned}
\tag{12}
$$

where for the first equality we used equation (11) and for the third the fact that $x$ and $\epsilon$ are almost surely uniquely determined by $\tilde{x}$.

Both probability measures on the right-hand side have a density. For $\mathbb{P}_X$ with respect to $dV_f$, see Section 2, this density is $p^*(x)$. Similarly, since $N_x$ is a linear subspace of $\mathbb{R}^D$, $q_n(\tilde{x}|x)$ is the density of $\mathbb{P}_{\tilde{X}|X=x}$ with respect to the volume form $dV_h$ where $h$ is the mapping from $\mathbb{R}^{D-d}$ to $N_x$, see Remark 1.

Then, the corresponding density of $\mathbb{P}_{\tilde{X}}$ is with respect to the product measure $dV_h \cdot dV_f$. However, this product measure is equivalent to $\lambda_D$ when restricted to subsets of $\widetilde{\mathcal{X}}$ and thus $q_n$ is the density of $\mathbb{P}_{\tilde{X}}$ with respect to $\lambda_D$.[5] Therefore, we can write equation (12) in terms of densities as follows:

$$
q_n(\tilde{x}) = p^*(x)q_n(\tilde{x}|x)
\tag{13}
$$

and it holds that

$$
\begin{aligned}
\int_{\widetilde{\mathcal{X}}} q_n(\tilde{x})d\tilde{x} &= \int_{\mathcal{X}} \int_{N_x} p^*(x)q_n(\tilde{x}|x)dV_h(\tilde{x})dV_f(x) \\
&= \int_{\mathcal{X}} p^*(x)dV_f(x) \\
&= 1,
\end{aligned}
\tag{14}
$$

as needed for a density on $\widetilde{\mathcal{X}}$. By setting $\tilde{x}$ to $x$ in equation (13), we obtain equation (7). This ends the proof.

**Remark 2** *Note that in Theorem 1, we need that $\tilde{X}$ is diffeomorphic to $\mathbb{R}^D$. This requires that the noise distribution $q_n(\cdot|x)$ is continuous for all $x$.*

### A.2 PROOF OF PROPOSITION 1

The generating function $f$ is an embedding for $\mathcal{X}$ and $X = f(Z)$ has the density $p^*(x)$ for $x \in \mathcal{X}$. We may extend the domain of $p^*(x)$ to include all points $x \in \mathbb{R}^D$ using the Dirac-delta function as follows

$$
p^*(x) = \int_{\mathcal{Z}} \delta(x - f(z))\pi(z)dz
\tag{15}
$$

After inflating $X$, we have that

$$
p_\Sigma(\tilde{x}) = \int_{\mathcal{Z}} \mathcal{N}(\tilde{x}; f(z), \Sigma)\pi(z)dz
\tag{16}
$$

with covariance matrix $\Sigma \in \mathbb{R}^{D \times D}$. Assume $\tilde{x} = x$ for some $x \in \mathcal{X}$. We Taylor expand $f(x)$ around $z_0 = f^{-1}(x)$ up to first order,

$$
f(z) \approx f(z_0) + J_f(z_0)(z - z_0),
\tag{17}
$$

and $\pi(z)$ up to second order,

$$
\pi(z) \approx \pi(z_0) + \pi(z_0)'(z - z_0) + \frac{1}{2}(z - z_0)^T \pi''(z_0)(z - z_0).
\tag{18}
$$

---

[5]This is because the column vectors of $J_f$ and $J_h$ form a basis of $\mathbb{R}^D$.

where $\pi(z_0)'$ denotes the gradient and $\pi''(z_0)$ the Hessian of $\pi$ evaluated at $z_0$, thus $\pi(z_0)' \in \mathbb{R}^d$ and $\pi''(z_0) \in \mathbb{R}^{d \times d}$. Then, we can approximate $p_\Sigma(x)$ as follows:

$$p_\Sigma(x) \approx \frac{1}{\sqrt{(2\pi)^D \det(\Sigma)}} \int_\mathcal{Z} \exp(-\frac{1}{2}(z - z_0)^T J_f^T \Sigma^{-1} J_f(z - z_0)) \cdot$$

$$\cdot (\pi(z_0) + \pi(z_0)'^T(z - z_0) + \frac{1}{2}(z - z_0)^T \pi''(z_0)(z - z_0))dz. \tag{19}$$

Now define $\hat{\Sigma}^{-1} = J_f^T \Sigma^{-1} J_f$. Then,

$$p_\Sigma(x) \approx \frac{\sqrt{\det(\hat{\Sigma})}}{\sqrt{(2\pi)^{D-d} \det(\Sigma)}} \int_\mathcal{Z} \frac{1}{\sqrt{(2\pi)^d \det(\hat{\Sigma})}} \exp(-\frac{1}{2}(z - z_0)^T \hat{\Sigma}^{-1}(z - z_0)) \cdot$$

$$\cdot (\pi(z_0) + \pi(z_0)'^T(z - z_0) + \frac{1}{2}(z - z_0)^T \pi''(z_0)(z - z_0))dz. \tag{20}$$

Thus, we can exploit this Gaussian in $\mathcal{Z}$-space and get

$$p_\Sigma(x) \approx \frac{\sqrt{\det(\hat{\Sigma})}}{\sqrt{\det(\Sigma)}}(\pi(z_0) + \frac{1}{2}\mathbb{E}\left[(z - z_0)^T \pi''(z_0)(z - z_0)\right])$$

$$= \frac{\sqrt{\det(\hat{\Sigma})}}{\sqrt{\det(\Sigma)}}(\pi(z_0) + \frac{1}{2}||\pi''(z_0) \odot \hat{\Sigma}||_+), \tag{21}$$

where $\odot$ stands for the elementwise multiplication and $||A||_+ = \sum_{i,j=1}^d A_{ij}$ for a $\mathbb{R}^d \times \mathbb{R}^d$ matrix $A$.

For the special case where $\Sigma = \sigma^2 I_D$, we can simplify this expression by exploiting that

$$\frac{\sqrt{\det(\hat{\Sigma})}}{\sqrt{(2\pi)^{D-d} \det(\Sigma)}} = \frac{1}{(2\pi)^{\frac{D-d}{2}}} \frac{\sigma^{-D}}{\sigma^{-d}\sqrt{g^f}}$$

$$= \frac{1}{(2\pi\sigma^2)^{\frac{D-d}{2}}\sqrt{g^f}}. \tag{22}$$

Thus, in total, we get for this special choice of $\Sigma$

$$p_\sigma(x) \approx \frac{1}{(2\pi\sigma^2)^{\frac{D-d}{2}}\sqrt{g^f}}(\pi(z_0) + \frac{\sigma^2}{2}||\pi''(z_0) \odot (J_f^T J_f)^{-1}||_+)$$

$$= \frac{1}{(2\pi\sigma^2)^{\frac{D-d}{2}}\sqrt{g^f}}\pi(z_0)(1 + \frac{\sigma^2}{2\pi(z_0)}||\pi''(z_0) \odot (J_f^T J_f)^{-1}||_+) \tag{23}$$

We assume now

$$\left|\frac{\sigma^2}{2\pi(z_0)}||\pi''(z_0) \odot (J_f^T J_f)^{-1}||_+\right| \ll 1. \tag{24}$$

Note that $1/(2\pi\sigma^2)^{\frac{D-d}{2}}$ from equation (23) is exactly the normalization constant obtained when inflating the manifold with Gaussian noise in the normal space, $q_n(x|x) = 1/(2\pi\sigma^2)^{\frac{D-d}{2}}$. What follows is that $\lim_{\sigma^2 \to 0} p_\sigma(x)/q_n(x|x) = p^*(x)$ as we wanted to show.

## A.3 PROOF OF THEOREM 2

The theorem follows directly from the definition of the reach number $\tau_\mathcal{X}$ of $\mathcal{X}$. It is defined as the supremum of all $r \geq 0$ such that the orthogonal projection $\text{pr}_\mathcal{X}$ on $\mathcal{X}$ is well-defined on the $r$−neighbourhood $\mathcal{X}^r$ of $\mathcal{X}$,

$$\mathcal{X}^r := \{\tilde{x} \in \mathbb{R}^D | \text{dist}(\tilde{x}, \mathcal{X}) \leq r\} \tag{25}$$

where $\mathrm{dist}(\tilde{x}, \mathcal{X})$ denotes the distance of $\tilde{x}$ to $\mathcal{X}$. Thus,

$$\tau_{\mathcal{X}} = \sup\left\{r \geq 0 \mid \forall \tilde{x} \in \mathbb{R}^D, \ \mathrm{dist}(\tilde{x}, \mathcal{X}) \leq r \implies \exists! x \in \mathcal{X} \text{ s.t. } \mathrm{dist}(\tilde{x}, \mathcal{X}) = ||\tilde{x} - x||\right\}, \quad (26)$$

see Definition 2.1. in Berenfeld & Hoffmann (2019). By assumption $\tau_{\mathcal{X}} > 0$. Thus for all $\tilde{x} \in \mathcal{X}^{\tau_{\mathcal{X}}}$ we have that $x := \mathrm{pr}_{\mathcal{X}}(\tilde{x})$ is unique. Since $\mathcal{X}$ is a manifold, it must hold that $\tilde{x} \in N_x$ where $N_x$ denotes the normal space in $x$. Let the noise generating distributions be a uniform distribution on the ball with radius $\tau_{\mathcal{X}}$, thus

$$q_{\mathrm{n}}(\tilde{x}|x) = \mathrm{Uniform}(\tilde{x}; B(x, \tau_{\mathcal{X}}) \cap N_x), \quad (27)$$

where $B(x, \tau_{\mathcal{X}})$ denotes a $D-$dimensional ball with radius $\tau_{\mathcal{X}}$ and center $x$.. Then, we have for $\widetilde{\mathcal{X}} = \bigcup_{x \in \mathcal{X}} N_{q_{\mathrm{n}}(\cdot|x)}$ that

$$\widetilde{\mathcal{X}} = \mathcal{X}^{\tau_{\mathcal{X}}}. \quad (28)$$

Thus, $\mathcal{X}$ is $Q-$normally separable where $Q := \{q_{\mathrm{n}}(\cdot|x)\}_{x \in \mathcal{X}}$.

# B  EXPERIMENTS

## B.1  TECHNICAL DETAILS FOR CIRCLE EXPERIMENTS

For the Normalizing Flow $T^{-1}(x)$ we use a BNAF (Block Neural Autoregressive Flow) for the circle experiment. The number of hidden dimensions was adapted to the dimensionality of the data and the difficulty of the target density. These details are reported in the corresponding tabular. For the optimization scheme, we used Adam optimizer with an initial learning rate 0.1, a learning rate decay of 0.5 after 2000 optimization steps without improvement (learning rate patience). The batch size was set to 200. The total number of iterations (one iteration corresponds to updating the parameters using one batch sample) used is also reported in the tabular. No hyperparameter fine-tuning was done.

For the FOM and $\chi^2-$noise models, we use the same architecture as for the $D = 2$ case.

| Data dimension | hidden layers | hidden dimension | total parameters | iterations |
|:---:|:---:|:---:|:---:|:---:|
| 2 | 3 | 100 | 31,204 | 70000 |
| 5 | 3 | 250 | 192,010 | 70000 |
| 10 | 3 | 500 | 764,000 | 70000 |
| 15 | 3 | 750 | 1,716,030 | 100000 |
| 20 | 3 | 1000 | 3,048,040 | 100000 |

Table 1: BNAF details for circle experiments.

## B.2  TECHNICAL DETAILS FOR MIXTURE OF VON MISE DISTRIBUTIONS ON $S^2$

For the Normalizing Flow $T^{-1}(x)$ we use rational-quadratic neural spline flows, alternating coupling layers and random feature permutations, see Durkan et al. (2019). For the optimization scheme, we used AdamW optimizer with an initial learning rate 0.0003, a learning rate cosine decay to 0 after every 2000 optimization steps, see Loshchilov & Hutter (2016), a weight decay of 0.0001 and a dropout probability of 0. The batch size was set to 200. No hyperparameter fine-tuning was done. See table 2 for more details. We use the same architecture for the FOM model and 3 seeds for the error bars.

| Coupling | residual blocks | hidden features | bins | spline range | total parameters | iterations |
|:---:|:---:|:---:|:---:|:---:|:---:|:---:|
| 10 | 3 | 50 | 8 | 3 | 171,845 | 50000 |

Table 2: BNAF details for a mixture of von Mises on $S^2$.

The target distribution is a mixture of four von Mises distributions, $p_1^*(\phi_1, \theta_1), p_2^*(\phi_2, \theta_2), p_3^*(\phi_3, \theta_3)$ and $p_4^*(\phi_4, \theta_4)$. Each of those distributions has the same product form

$$p_i^*(\phi_i, \theta_i) = \frac{\exp(\kappa \cos(\theta_i - \mu_i))}{2\pi I_0(\kappa)} \cdot \frac{\exp(\kappa \cos(2(\phi_i - m_i)))}{\pi I_0(\kappa)}. \tag{29}$$

We set $\kappa = 6$. However, they differ in their mean values $\mu_i$ and $m_i$, see table 3. We used 3 different seeds in total to obtain the confidence intervals.

| i | $\mu_i$ | $m_i$ |
|---|---------|-------|
| 1 | $\frac{\pi}{2}$ | $\frac{\pi}{4}$ |
| 2 | $\frac{4\pi}{3}$ | $\frac{3\pi}{4}$ |
| 3 | $\frac{\pi}{2}$ | $\frac{3\pi}{4}$ |
| 4 | $\frac{\pi}{4}$ | $\frac{4\pi}{3}$ |

Table 3: Mean values for the mixture of von Mises distributions.

