# OpenReview forum: "Density estimation on low-dimensional manifolds: an inflation-deflation approach"
_ICLR.cc/2021/Conference — Reject_

### Official Review · AnonReviewer2 · 2020-10-19
**Nice theoretical work**

**Rating:** 7
**Confidence:** 2

**Review:**

In this paper, the authors address main limitations of Normalizing Flows (NFs) method for estimation of density functions on manifolds. Since NFs requires the support of density function to cover the whole Euclidean space, they propose to add noise (inflate) to apply NF. The paper is nicely written with clear introduction of basic concepts very useful for non-expert readers. They provide theoretical guarantees on the variance and type of added noise that make the method work and illustrate with synthetic experiments.
My only concern is about the applicability and usefulness on real data and specific machine learning. I have found some issues and also have some questions. See below:
Major issues:
-	How this method can be used for a real problem with real datasets. Could you please provide some example or give some guidance about the potential of this method for the machine learning community?
-	It is shown that when D>>d, Gaussian noise is an excellent approximation of noise restricted in the normal direction and the experiments seem to confirm this. What should be the criterion to say that D is large enough to make this approximation useful? Can you give some way to test it in practice?
-	Example 2: a graphical description of this example would help, for example, showing that variance of noise can help to make the Q-normally separated condition to be met.
-	Page 5: I didn’t understand the following sentence “A potential future avenue is to \sigma^2A and this reach number”. Could you please elaborate it?

Minor issues
-	In Example 1: “an z” -> “a z”
-	References have missing information. For example, for Cornish et al, Gemici et al and Rezende et al papers, there is not information about the source. Did they works published in conferences, journals or ArXiV?
-	Page 12: Please correct the number of table in “Table ??”

---

> ### Author Response · Authors · 2020-11-25
> **Response to AnonReviewer2**
>
> We thank the reviewer for reading our manuscript and for raising many interesting questions. We are happy that she or he appreciated our work and thank the reviewer for spotting some minor issues. We corrected them in the new manuscript.
>
> We thank the reviewer for raising the concern for real datasets. We think our method can be useful in various ways and added a new paragraph in the discussion section, page 9, to address this question.
>
> Regarding the reviewer's interesting question when $D$ is big enough compared to $d$ in order to  approximate Gaussian noise in the normal space with vanilla Gaussian noise, equation (8) serves as a rule of thumb. For instance, for MNIST we have that $D=784$ and for most digits that $d\approx10$, see e.g. [1]. Thus, according to equation (8), we make an expected squared error of $10/772 \approx 0.013$ when approximating Gaussian noise in the normal component with full Gaussian noise. To fully address this question is beyond the scope of equation (8) as the quality of approximation highly depends on the manifold's curvature $f''$ and on $\pi''(z)$ (where $\pi(z)$ is the generating latent distribution).
>
> We thank the reviewer for proposing a graphical description of $Q$ normal separability in Example 1. We agree that it needs some time to understand the Q-normal separability and a graphical description would help to do so. We, therefore, added a new figure (Figure 2 on page 4) which aims to help the understanding of Example 2 and the concept of Q-normal separability.
>
> Finally, we apologize for the confusion regarding the connection to the reach number in the last sentence of Section 3. There was a word missing. However, we elaborated on this connection and formulated a new Theorem (Theorem 2 on page 5). There, we establish a connection to the reach number.
>
> References:
>
> [1] Matthias Hein and Jean-Yves Audibert. Intrinsic dimensionality estimation of submanifolds in rd.
> In Proceedings of the 22nd international conference on Machine learning, pp. 289–296, 2005.

---

### Official Review · AnonReviewer3 · 2020-10-23
**Interesting but not enough comparison with related works**

**Rating:** 6
**Confidence:** 3

**Review:**

This paper proposes a method for estimating the probability deinsity distribution on a low-dimensional manifold embedded in a high-dimensional space using Normalizing Flow (NF). The problem is that the universality of NF is limited because low-dimensional manifolds are not diffeomorphic with respect to high-dimensional Euclidean space. The proposal of this study is to make NF applicable by inflating low-dimensional manifolds with Gaussian noise. Then, after the transformation is obtained, the probability distribution on the original low-dimensional manifold can be obtained by deflation.

There is a detailed discussion about the addition of Gaussian noise. Theoretically, any Gaussian noise $ e $ at coordinate x can be decomposed into projections on tangent space $ T_x $ and normal space $ N_x $ as $ e = e_t + e_n $. Only normal noise $ e_n $ is ideal to add. However, it is not realistic to find this at each coordinate x. This paper argues that Gaussian noise $ e $ is a good approximation of $ e_n $ when the manifold dimension $ d $ is sufficiently smaller than the higher-dimensional Euclidean space dimension $ D $. This also argues that the noise variance $ \ sigma $ at that time should be set according to the inverse of the density.

In computer experiments, simple simulation results using the von mises distribution on a circle and a sphere are shown.

The gist of the manuscript is well-written, and the issues it deals with are also important and interesting.
Theoretically, interesting discussions are being developed, but discussions and experimental results are somewhat weak regarding the merits of practical application.

I have some questions.
1. I couldn't understand the description about scalability well, "our method scales to high dimensions because it is based on one NF and does not require to compute its inverse." Isn't the scalability the same as normal NF because the proposed method basically uses normal NF?

2. In this research, Authors propose to add noise to the data sampled from low-dimensional manifolds in a pseudo manner, but in actual measurement, noise has been included in data naturally without adding it in a pseudo manner. Is it necessary to add pseudo noise even when applying it to real world data?

3. The denoising auto-encoder is famous as a manifold learning method that adds pseudo noise. Is there any relation to this? Also, I would like to know if there are any advantages of the proposed method over the denoising auto-encoder.

4. VAE is a well-known method for finding the mapping of low-dimensional space to the normal distribution. Is there any relation to this? Also, I would like to know if there are any advantages of the proposed method over VAE.

5. In relation to the above, it may be good to have a comparison experiment with denoising auto-encoder and VAE.

6. The lack of experimental results with actual data makes that the paper is unconvincing. Especially for the description of scalability, it is better to prove it experimentally.

7. This is a pure question, is it possible to know the dimensions of the manifold through this technique?

---

> ### Author Response · Authors · 2020-11-25
> **Response to AnonReviewer3**
>
> We thank the reviewer for reading our manuscript carefully and attentively. The reviewer's comments helped us a lot to better pigeonhole our method into the literature.
>
> Indeed, the scalability of our method is the same as for NFs. We agree that the description of scalability as it was stated at the end of Section 1, page 1, and in the abstract is misleading.  The comment only refers to the necessity to invert the flow as it is needed in e.g.  [1]. This inversion adds additional complexity depending on which NF one is using. E.g. using an Autoregressive flow, the forward computation can be done in parallel. However, the inverse computation is $D$-times slower as computations can't be done in parallel anymore, see chapter 3.1 in [2].
> As a consequence, we decided to update the abstract and the sentence at the end of Section 1 (page 1).
>
> We thank the reviewer for the curious question on the nature of noise we need for the inflation step. Actually, the manifold hypothesis states that the data-cloud is only living close to a manifold. If the real noise is sufficiently large, then we don't need our method and a usual NF can be used to learn $p^*(x)$. If the noise is too small (which practically manifests in a degeneration of the flow's log Jacobian determinant), we should add synthetic noise.
>
> We thank the reviewer to mention VAE and denoising auto-encoder (DAE). Indeed, both methods have some similarities to ours. We touch on these similarities before we explain the differences.
>
> The objective of DAE is to recover a noise-corrupted signal, as it is the case for our method. However, the DAE learns to recover the data-point $x$ from the noisy one $\tilde{x}$, whereas in our method we recover the likelihood $p^*(x)$ from the inflated one $q_{n}(\tilde{x})$. DAE does not estimate the actual density. Nevertheless, one could use some theoretical insights from the DAE literature to estimate the density. This was done in [3]. The drawback here is that the density is only learned up to a normalization constant which needs to be estimated using Monte-Carlo integration and therefore suffers from the curse of dimensionality.
>
> VAE also make use of the fact that data lives on a (noisy-)manifold by learning a probabilistic encoder and decoder. However, the objective function is a lower bound on the data log-likelihood and not the log-likelihood $\log p^*(x)$ itself. It is therefore not a surprise that VAE has difficulties estimating $p^*(x)$ despite tighter bounds, see [4]. Moreover, in VAE the probabilistic decoder is assumed to be a Gaussian distribution which implicitly makes $p^*(x)$ a density  supported on $\mathbb{R}^D$. Hence, to our understanding, learning exact densities with respect to a Riemannian volume form is beyond the scope of VAE.
> Nevertheless, our theory can be used to learn low-dimensional representations (as VAE can do). We added a new paragraph in the Discussion section where we discuss potential future research based on our theory.
>
> Altogether, we think that a comparison to VAE or DAE is not appropriate for the aim of this paper (estimating the exact density function supported on a low-dimensional manifold).
>
> Finally, we thank the reviewer for the very interesting question if we can use our method to learn $d$. This question will be kept for further work.
>
> References:
>
> [1] Johann Brehmer and Kyle Cranmer. Flows for simultaneous manifold learning and density estimation.
> arXiv preprint arXiv:2003.13913, 2020.
>
> [2] George Papamakarios, Eric Nalisnick, Danilo Jimenez Rezende, Shakir Mohamed, and Balaji Lakshminarayanan.
> Normalizing flows for probabilistic modeling and inference. arXiv preprint
> arXiv:1912.02762, 2019.
>
> [3] Siavash A Bigdeli, Geng Lin, Tiziano Portenier, L Andrea Dunbar, and Matthias Zwicker. Learning
> generative models using denoising density estimators. arXiv preprint arXiv:2001.02728, 2020.
>
> [4] Tom Rainforth, Adam R Kosiorek, Tuan Anh Le, Chris J Maddison, Maximilian Igl, Frank Wood,
> and Yee Whye Teh. Tighter variational bounds are not necessarily better. arXiv preprint
> arXiv:1802.04537, 2018.

---

### Official Review · AnonReviewer4 · 2020-10-24
**Mathematical part is vague, no experiments**

**Rating:** 5
**Confidence:** 4

**Review:**

The paper tackles the problem of recovering a probability distribution, which is supported in a low-dimensional manifold. When the dimension of that manifold is full, the problem can be solved by the Normalizing Flow method.

Since the dimension is not full, it is suggested to add a gaussian noise to the data points (this is equivalent to a certain convolution of the initial distribution function, ie equation 3). Then full dimensional manifold is recovered and subsequently deconvolution is applied (equation 7).

Certain mathematical aspects of the narrative are vague. Eg \tilde{X} is defined as X x Noise, but then in equations 4 and 5, it becomes a subset of the plane.

The notion of Q-normally separated is never satisfied in practice. It is only slightly stronger than "all the normal spaces have no intersections at all". The deconvolution made by equation 7 is rarely is a good solution.

Also, the experimental part is purely dealing with synthetic examples. This could be because the deconvolution by equation 7 is not the best way to recover the signal.

It is claimed that the first goal of synthetic experiments is to confirm the scaling factor in equation (7) numerically. But experiments deal with highly specific synthetic data - a curve (ie, 1D manifold) on the plane. It is doubtful that such a simple experimental framework can justify eq (7). It is not computationally difficult to check the quality of deconvolution in a style of eq (7) for higher-dimensional data.

---

> ### Author Response · Authors · 2020-11-25
> **Response to AnonReviewer4**
>
> We thank the reviewer for reading our manuscript carefully and helping us to improve our theory. We agree that certain mathematical aspects were not precise enough. For instance, equations (4) and (5) were in conflict with the definition of $\tilde{X}$ as a Cartesian product. We, therefore, updated the Definition $\tilde{X}$. Now it is a union of sets in $\mathbb{R}^D$ and we changed Example 1 accordingly, see page 3. We also found that the noise distribution was used with some abuse of notation and therefore added Remark 1 to clarify the role of it.
>
> We thank the reviewer for sharing the concern regarding the Q-normal separability. We have shown that there exist a set of noise distributions such that $Q$-normal separability is fulfilled for spheres  (and it is easy to show that this is true for tori as well). There are many examples for real-world data-sets on spheres, see e.g. [1] or [2]. Hence, for these datasets, the Q-normal separability is fulfilled using the appropriate noise.
>
> To address more complex manifolds, we added a new theorem in Section 3.3. page 6, which provides a sufficient condition such that a manifold is Q-normally separable. Essentially, the condition is that the manifold's reach number is positive. Hence, if the manifold is sufficiently regular and has a positive reach number, we can find a collection of normal space noises such that the manifold is $Q$-normally separable. This includes a wide range of manifolds.
>
> Finally, we want to emphasize that Q-normal separability is not necessary in practice to obtain good estimates, as our experiments have shown.
>
> These experiments indeed deal with specific synthetic data. Most of the literature in manifold learning on low-dimensional manifolds is dealing with easy manifolds such as circles, spheres, or tori. Therefore, we decided to benchmark our performance on such examples as well.
>
> We disagree with the reviewer's opinion that we did not confirm the scaling factor in equation (7). In fact, we confirmed it in various ways.
>
> - For the normal Gaussian noise, each $\sigma^2$ corresponds to a different scaling factor, namely $\sqrt{2\pi \sigma^2}$. The flat course of the Kolmogorov-Smirnov (KS) statistics in figure 3 (top left) is a strong indicator that the factor is correct.
>
> - We embedded the circle into higher dimensions. For $D=5,10,15,20$ we iterated the experiment and evaluated the KS statistics. Note that the scaling factor depends on $D$, namely $(2\pi \sigma^2)^{(D-d)/2}$ where $d=1$ for the circle. We displayed the course of the KS-statistics for $D=20$ in figure 3 (top right) but the plots for $D=5,10,15$ look very similar (see, for instance, the lower-left image in figure 3 which shows the best KS value depending on $D$). The fact that these courses are very similar to the one obtained for $D=2$, in particular the flat plateau, is also a strong indicator that the scaling factor is correct. We added a new footnote (footnote 4 on page 6) to emphasize this.
>
> - We also considered a sphere. For this 2D manifold, the course of the KS-statistics in figure 4 (lower right) suggests that the scaling factor is indeed correct.
>
> - We inflated the manifold with $\chi^2$ noise. Although the inflation looks very similar to the one obtained with normal Gaussian noise, only the normalization constant of the $\chi^2$ distribution is the correct scaling factor.
> Of course, we can find a $\sigma^2$ such that the Gaussian noise scaling factor is equal to the one for the $\chi^2$-noise, namely $\sigma^2\approx 6.7$. However, normal Gaussian noise performs very poor with such a magnitude. This confirms that indeed the normalization constant of the noise used to inflate the manifold is the correct one as predicted by equation (7).
> \end{enumerate}
> References:
>
> [1] Karpatne, A., Ebert-Uphoff, I., Ravela, S., Babaie, H. A., and Kumar, V. (2019). Machine learningfor the geosciences: Challenges and opportunities.IEEE Transactions on Knowledge and DataEngineering, 31(8):1544–1554.
>
> [2] Hamelryck, T., Kent, J. T., and Krogh, A. (2006). Sampling Realistic Protein Conformations UsingLocal Structural Bias.PLoS Computational Biology, 2(9).

---

### Official Review · AnonReviewer1 · 2020-10-28
**Interesting framework to train NF for manifold-supported data. Questions about theoretical development and practical utility.**

**Rating:** 6
**Confidence:** 3

**Review:**

This work presents a novel theoretical development to tackle the problem of estimating normalising flows (NF) for data with support on complex manifolds. Motivated by the fact that NF are diffeomorphic  transformations from a simple space, ideally Euclidean, the author address the problem of modeling data which distribution is defined on more complex and unknow manifolds.
The idea consists in inflating the data manifold with suitable noise (normal noise) in order to make it diffeomorphic to a simpler space where a NF can be estimated. The data density can be therefore approximated from the inflated space through an opportune deflation operation.

The main contribution of the work consists in formalising the conditions on the inflating noise to obtain the results of Theorem 1. In practice, Proposition 1 shows that the noise can be drawn from a Gaussian with variance depending from the curvature of the manifold. The experimental section demonstrates the theory on simple synthetic cases on S2.

The idea proposed in the paper is interesting, and of potentially utility. The derivation is however unclear in some aspects, especially concerning the definition of Q-separation. In particular, Definition 1 implies null measure for the set of manifold points spanning the normal subspace. However, to my understanding, this definition is not sufficient to guarantee the unicity of the mapping between normal space and manifold point. For example, any finite collection of manifold points would still have zero measure. In this case however, the unique correspondence between normal space and manifold is broken. Back to Example 2, this non-unicity can be achieved by increasing the noise along the normal direction to cross the origin (0,0). In this case each point of the inflated space has two generators, making the isomorphic mapping  impossible. I can understand that this aspect is related to the choice of the noise discussed in Section 3.3, but this issue still undermines the conclusion of Theorem 1. I would encourage the authors to clarify this point.

There are  further technical aspects deserving clarification. In Example 2, the definition of the set A(\tilde{x}) should not contemplate the empty set. Rather, A(\tilde{x}) should be the point x’ if \tilde{x} is not (0,0). Moreover, in Proof A.1, equation (13), the relation between the q_n and \hat{q}_n in the two first integrals, and the reason why they can be exchanged, is not clear.

Finally, some questions arise concerning the practical feasibility of the proposed theory. Computing the curvature of the data manifold may be extremely challenging in presence of complex data, especially in the low sample-size regime. In this case, the estimation of the noise parameter in formula (9) may be practically impossible.

---

> ### Author Response · Authors · 2020-11-25
> **Response to AnonReviewer1**
>
> We thank the reviewer for reading the manuscript carefully, making many useful comments, and spotting a fundamental error. Indeed, the definition as stated was too weak. Every point in the inflated manifold must have an a.s.-unique generator as mentioned in the preface of Section 3.2. We apologize for this mistake. We updated Definition 1 accordingly.
>
> With the corrected definition of $Q-$normal separability having 2 generators for a single point $\tilde{x}$ is not possible anymore. We updated the example according to the new definition and corrected the wrongly used empty set (we thank the reviewer for spotting this mistake). To clarify the concept of Q-normal separability, we added a new figure for example 2 on page 4.
>
> However, it is worth to remark that even though a unicity of the generator is required,  if the probabilities of the other generators are sufficiently low, the Q-normal separability is fulfilled in practice.
>
> We took the reviewer's concern about the validity of the proof very seriously and decided to make it mathematically more rigorous. We also added a new Remark on page 3 to clarify the role of the noise distribution $q_{\rm{n}}(\tilde{x}|x)$.
>
> We thank the reviewer for the remark on the practical feasibility. We agree that it is challenging to use formula (9) in practice for complex data manifolds. However, our experiments suggest that not being able to compute the upper bound is not a big problem. The range for $\sigma^2$ leading to a good performance is wide. Also, we note that our theory can always be used as an approximation  for $p^*(x)$ using any value of $\sigma^2$.
> In addition, our new Theorem 2 provides an interesting connection to the so-called reach number from the manifold learning literature for which estimates exist, see [1].
>
> Finally, to address the practical usage of our theory, we added a new paragraph in the discussion section.
>
> Further relevant modifications of our Submission (not mentioned in any of the answers below):
>
> - Equation (3) is not the convolution with the noise in the normal space anymore. Previously $q_{\rm{n}}(\tilde{x}|x)$ was not defined for $\tilde{x}\notin N_x$ and therefore the convolution was ill-posed.
>
> - We changed the name of the variable in Example 1 from $\alpha$ to $\gamma$ to make it consistent with our new Remark 1.
>
> - Equation (8), the expected relative squared error was wrongly denoted by $\mathbb{V}$ rather than $\mathbb{E}$
>
> - We modified the introduction in example 4.1 to clarify that the Flow on manifolds (FOM) method refers to the method proposed in [2].
>
> - Figure 5, we added the Gaussian in the normal space in the KS-performance plot (lower right) for consistency reasons. Also, we used more expressive Flows which is the reason why the performance for all models slightly increased.
>
> - We removed Remark 1 on page 11.
>
>
> References:
>
> [1] Eddie Aamari, Jisu Kim, Frédéric Chazal, Bertrand Michel, Alessandro Rinaldo, Larry Wasserman,
> et al. Estimating the reach of a manifold. Electronic journal of statistics, 13(1):1359–1399, 2019.
>
> [2] Mevlana C Gemici, Danilo Rezende, and Shakir Mohamed. Normalizing flows on riemannian manifolds.
> arXiv preprint arXiv:1611.02304, 2016.

---

### Decision · Program_Chairs · 2021-01-07
**Final Decision**

**Decision:**

Reject

**Comment:**

The paper provides an interesting set of theoretical ideas to improve the estimation of normalizing flows on datasets that fail to be fully dimensional. Although the method is appealing, I believe the paper falls a bit short of acceptance at the conference. Too many practical issues are left out, as discussed by reviewers, and the method seems promising but not fully connected to the rest of the literature on estimating low-dimensional distributions living in high dimensional spaces. We encourage the authors to use the feedback contained in this round of reviews to improve their work.